# Linking Endoplasmic Reticular Stress and Alternative Splicing

**DOI:** 10.3390/ijms19123919

**Published:** 2018-12-07

**Authors:** Nolan T. Carew, Ashley M. Nelson, Zhitao Liang, Sage M. Smith, Christine Milcarek

**Affiliations:** School of Medicine, Department of Immunology, University of Pittsburgh, E1059 Biomedical Science Tower, Pittsburgh, PA 15261, USA; ntc8@pitt.edu (N.T.C.); amd257@pitt.edu (A.M.N.); ZHL80@pitt.edu (Z.L.); SMS253@pitt.edu (S.M.S.)

**Keywords:** B cells, RNA splicing, ER stress, unfolded protein response, RIDD

## Abstract

RNA splicing patterns in antibody-secreting cells are shaped by endoplasmic reticulum stress, *ELL2* (eleven-nineteen lysine-rich leukemia gene 2) induction, and changes in the levels of *snRNA*s. Endoplasmic reticulum stress induces the unfolded protein response comprising a highly conserved set of genes crucial for cell survival; among these is Ire1, whose auto-phosphorylation drives it to acquire a regulated mRNA decay activity. The mRNA-modifying function of phosphorylated Ire1 non-canonically splices Xbp1 mRNA and yet degrades other cellular mRNAs with related motifs. Naïve splenic B cells will activate Ire1 phosphorylation early on after lipopolysaccharide (LPS) stimulation, within 18 h; large-scale changes in mRNA content and splicing patterns result. Inhibition of the mRNA-degradation function of Ire1 is correlated with further differences in the splicing patterns and a reduction in the mRNA factors for snRNA transcription. Some of the >4000 splicing changes seen at 18 h after LPS stimulation persist into the late stages of antibody secretion, up to 72 h. Meanwhile some early splicing changes are supplanted by new splicing changes introduced by the up-regulation of ELL2, a transcription elongation factor. ELL2 is necessary for immunoglobulin secretion and does this by changing mRNA processing patterns of immunoglobulin heavy chain and >5000 other genes.

## 1. Introduction

In the transition from naïve B lymphocytes to antibody-secreting cells, a characteristic morphological change is the appearance of a large amount of the endoplasmic reticulum (ER) along with the production of large quantities of the secreted form of the immunoglobulin (Ig) protein; these activated B cells can persist and secrete Ig for a long time [1,2]. Similar large-scale changes in the ER occur when pancreatic beta cells are induced to secrete insulin and when liver cells secrete bile or other soluble products [3]. Understanding the enormous changes the cells go through in forming the ER has engrossed researchers for years. 

The ER is composed of stacks of membranes separated by lumen where proteins synthesized for cellular export are made (rough ER) or where membrane lipids are produced (smooth ER). The lumen of the ER has a very high calcium (Ca^++^) ion concentration and is an oxidative environment in contrast to the cytoplasm. This facilitates proper protein folding, aided in large part by molecular chaperones. Beyond facilitating proper folding, some ER chaperone proteins can stabilize their “client” proteins and prevent ubiquitination which might lead to client degradation [4]. BiP (binding immunoglobulin protein) was first discovered as a chaperone for immunoglobulin heavy chain (IgH) in developing B cells; it was subsequently described as binding IgH until it could associate with the light chain [5]. It has been renamed *Grp78* or *Hspa5* and is a major player in activating Ire1 (inositol-requiring enzyme 1) and other proteins in the ER stress responses. Localized in the lumen of the endoplasmic reticulum, Grp78 is involved in the folding and assembly of many proteins in the ER. There are many members of the ER chaperone protein family some involved in each step of protein folding, degradation and autophagy pathways. The action of these ER chaperone proteins in neurodegenerative diseases is extensively reviewed in a recent article [6].

In cells that differentiate to produce large amounts of secreted proteins, the ER expands rapidly upon certain predetermined signals. Stress from disturbances in redox regulation, glucose deprivation, viral infection, or a back-up in the proteasome function can lead to excess protein as well; these all can contribute to the unfolded protein response (UPR), which is a highly conserved set of proteins seen throughout evolution. For example, 18–24 h after treating naïve B cells with lipopolysaccharide (LPS) there is minimal induction of Ig secretory-specific protein but the cells are beginning to enter the S phase from their previous quiescent state, their redox balance is altered, the amount of cytoplasmic and mitochondrial chaperones increases five-fold, Ire1 auto-phosphorylates, and Xbp1 is non-canonically spliced in the cytoplasm based on a defined stem loop and consensus sequence [7,8,9] to produce a transcription factor for UPR genes. The highly conserved ER stress-induced CHOP mRNA is up-regulated seven-fold [10]. CHOP is the DNA damage-inducible transcript 3, also known as C/EBP homologous protein, the induction of which causes calcium release from the endoplasmic reticulum into the cytoplasm, and if unchecked can result in apoptosis [11]. Calcium release may serve as an energy deprivation signal to stimulate mitochondrial ATP production and thereby compensate for the large amount of ATP consumed by protein folding in the ER [3]. Based on the severity of the ER stress, the UPR genes can re-establish normal ER function, sound an alarm and induce an adaptation to the higher levels of unfolded proteins, or result in cell death [1].

The UPR in many cells and across many phyla relies on three pathways: inositol-requiring enzyme (Ire1) and X-box protein (Xbp1); the activating transcription factor 6 (Atf6) pathway; and a third pathway involving the protein kinase R (PKR)-like endoplasmic reticulum kinase (Perk). All three pathways can be activated by factors that draw Grp78 away from its steady state location in the ER lumen where it is associated with Ire1, Atf6, and Perk to become involved in chaperoning the folding of nascent proteins. This allows Ire1 to auto-phosphorylate in the membrane and acquire the additional activities on RNAs described as regulated Ire1-dependent mRNA decay (RIDD). In mature antibody-secreting cells, the ER response is unique compared to that seen in other cells in that Ire1 phosphorylation and Xbp-1 activation have the largest effects on cell viability and antibody secretion. The Perk pathway in B cells is suppressed [12,13] and the Atf6 pathway has been shown to be dispensable [14]. This implies some unique features for B cells. Another unique feature of B cell activation is the fact that the naïve B cells utilize anaerobic glycolysis until the differentiation pathway with Prdm1 begins and switches the cells to oxidative metabolism in antibody-secreting cells, a more efficient source of ATP generation [15]. How this shift in metabolism interacts with the ER stress pathways is unknown but may provide intriguing links between metabolism and stress.

In this review, we discuss ER stress, and specifically the impact of RIDD on the pattern of mRNA expression observed. This process has not heretofore been extensively investigated. Namely, we will discuss the impact of RIDD on the alternatively spliced RNAs that result in B cells. The role of *ELL2*, a transcription elongation factor, induced later in the B cell development pathway toward antibody-secreting cells in directing alternative RNA splicing, will also be reviewed.

## 2. Model Systems for Studying ER Stress

### 2.1. B cell Activation to Antibody Secreting Cells

A schematic of the *in vitro* activation of naïve B cells, by bacterial lipopolysaccharide, to antibody-secreting cells (ASCs) is depicted in Figure 1. The time-line recapitulates the principal features of the activation of B cells that do not enter germinal centers in vivo. Within 18 to 24 h of stimulation, the Go resting cells leave their quiescent state and begin to proliferate and activate the ER stress response. This occurs prior to the massive up-regulation of production of the secreted form of the immunoglobulin molecule [16,17]. After Ire1 auto-phosphorylation by ER stress, the typical regulated Ire1-dependent mRNA decay pathway (RIDD) is activated, producing Xbp1 mRNA and degrading other cellular RNAs with a stem loop and consensus sequence [18]. When mice are deficient for Xbp1, B cell differentiation to antibody-secreting cells is blocked and the ER is distended [19,20,21,22]. In vitro, Ire1 cleaves the mRNA of secretory Ig μ chains (μs) in activated B cells. The IgM response is partially restored in Xbp1/Ire1 double knockout mice relative to the single Xbp1 knockout mice. RIDD can thus reduce Ig synthesis and secretion under certain circumstances [19].

Mammalian target of rapamycin (mTOR) encodes a kinase that promotes anabolic activities along with phosphorylation of an eIF4E binding protein and ribosomal S6 protein phosphorylation in many cells [23]. ER stress attenuates the mTOR pathway, reducing protein synthesis, while induction of ER stress in B cells leads to a de-phosphorylation of eIF2 alpha [21]. Treatment of B cells with rapamycin or by conditionally deleting RAPTOR, an essential signaling adaptor in the mTORC1 complex, blocked production of ASCs [24] in part by decreasing the expression of BiP, an unfolded protein chaperone. Artificial hyper-activation of the mTOR pathway occurs when the tuberous sclerosis complex (TSC1) is deleted in B cells; when B cells are engineered to over-activate the mTOR pathway there is a higher level of apoptosis following LPS stimulation [21]. When mice are conditionally deleted for both Xbp1 and TSC1, the typical Xbp1 distended ER is alleviated and Ig secretion is enhanced [20], indicating that mTOR can work at cross-purposes to Xbp1.

mTOR has also been shown to be upregulated in human marginal zone B cells, cells that contain more mitochondria, ER, lysosomes, and Golgi. Marginal zone B cells contrast with human follicular B cells that typically require T cell help for differentiation to secretion and have lower mTOR. The marginal zone B cells are more responsive to T cell independent stimuli and growth signals like TAC1 than the follicular B cells because mTOR links TAC1 signaling with metabolic events and differentiation [25].

### 2.2. Liver Secretion and the Effects of RIDD

Activation of the Ire1-mediated RIDD pathway in liver cells can lead to degradation of certain microRNAs and prevent hepatic steatosis, aka fatty liver [26,27]. Micro-array studies on mRNA survival identified RIDD target genes for lipogenesis and lipoprotein metabolism with the characteristic stem loop and consensus sequence [28]. In liver cells CReP/Ppp1r15b mRNA, encoding a regulatory subunit of eukaryotic translation initiation factor 2α (eIF2α) phosphatase, is a RIDD substrate. Decreased CReP expression results in more eIF2α phosphorylation and the attenuation of protein synthesis [29]. This is at odds with the result of less phosphorylated eIF2 alpha obtained in ER-stimulated B cells discussed above. This discrepancy should be explored.

Mice conditionally deficient in Xbp1 in the liver have profound hypo-lipidemia. The Xbp1 deficiency triggers abnormally high over-activation of Ire1 auto-phosphorylation and RIDD activity, in a feed-back mechanism. Loss of Ire1 phosphorylation was able to reverse the hypo-lipidemia resulting from Xbp1 deficiency in the whole animal [28].

### 2.3. Drosophila

A large number of mRNAs that are associated with the ER are degraded by RIDD in *Drosophila melanogaster* tissue culture cells [30]. Those degraded mRNAs contain a stem-loop structure like that in Xbp1. It was suggested that the mRNAs that were degraded are a specific subset, elimination of which could clear out the protein-folding machinery in anticipation of the production of new mRNAs. RIDD was also shown to be important in development of the eye in the whole *Drosophila* [31]. When the photoreceptors had Ire1 mutant forms there were defects in the delivery of rhodopsin-1 into the inter-rhabdomere space. The phenotype of an Xbp1 mutant photoreceptor did not have the same delivery defects indicating different roles for Ire1 and Xbp1.

### 2.4. Pancreatic Beta Cells

The mouse-derived MIN6, a cultured pancreatic beta cell line, recapitulates glucose-regulated pro-insulin biosynthesis and protective effects of NR4AI-induced resistance to ER stress-mediated apoptosis [32,33]. When MIN6 cells are treated with thapsigargin (TG) a drug known to induce ER stress within 2 to 5 h of treatment in the absence of any insulin synthesis [34], Ire1 and RIDD activation occur; there is an increase in the Xbp1 short/spliced RNA form, and ~five-fold CHOP (Ddit3) mRNA induction, all hallmarks of TG-induced stress [35]. These cells could serve as an excellent model system for the study of RIDD.

## 3. Ire1 Activation and ELL2 Induction with Alternative Splicing

### 3.1. ER Stress Activates RIDD 

In B cells, Ire1 auto-phosphorylation upon ER stress induces RIDD activity and non-canonical splicing on a unique stem loop in the Xbp1 mRNA leading to production of a transcription factor that induces a series of UPR genes [36]. The RIDD activity can lead to the destruction of a number of other mRNAs, including the immunoglobulin mRNA molecules under certain circumstances [19]. The RNAs that are subject to RIDD share the stem loop structure of the Xbp1, a consensus CUGCAG, but not the ability to be non-canonically spliced like Xbp1 [37]. It may be that modulation of mRNA destruction by the Ire1 RNase and the overall ER stress response can lead to divergent cell fates of death or survival [18,38]. Cells may use a “multi-tiered mechanism” that results in distinct outputs for Ire1-mediated RIDD [38]. Based on the crystal structure of the Ire1 dimer and its stabilization by ADP, it has been suggested that its activation is transient and subordinate to ER-luminal stress signals [39]. 

In mammalian cells, both RIDD and the non-canonical splicing of Xbp1 are observed with ER stress [8,9]. In budding yeast like *Sacchromyceses cerevisiae*, the Xbp1 equivalent HAC1 mRNA is stored in the cytoplasm but one special intron is removed, only under conditions of stress, and the mRNA is spliced in the cytoplasm using a tRNA ligase [40,41]. Splicing of HAC1 is not seen in *S. pombe*, fission yeast, and only RIDD is observed [42,43]. The RIDD involves ribosome stalling, and mRNA degradation in an mRNA no-go decay pathway [43]. In mammals, cleavage of Xbp1 and the RIDD activity have different requirements biochemically. Ire1 oligomeric-subunit interaction is required for maximal Xbp1 splicing while RIDD needs only dimer formation [44].

RIDD can be regulated by eliminating the Perk function in some cells [45] or eliminating a ribonuclease inhibitor (RNH1) found associated with it in cells [46]. A small molecule inhibitor of RIDD was found that blocks RNA substrate access to the active site of phosphorylated Ire1; this inhibitor is called 8-formyl-7-hydroxy-4-methylcoumarin/CB5305630 and is also known as 4u8C (4-methyl umbelliferon 8-carb-aldehyde) [47]. The drug does not interfere with CHOP activation but does block amylase secretion after dexa-methasome induction of mouse embryo fibroblasts. As we discuss below, 4u8C can influence mRNA splicing patterns.

### 3.2. Splicing Patterns Are Changed When B Cells Are Activated

We examined the splicing patterns of genes, comparing B cells and cells stimulated for 18–24 h with LPS when Ire1 is phosphorylated. We saw that more than 11,144 genes had alternative splice forms between the two conditions, shown in [48] and illustrated here in Figure 2. A set of about 4211 genes that changed splicing patterns early were seen to persist in cells at 72 h; we called these the ELL2 independent genes. When we inhibited the Ire1 phosphorylation and the RIDD function of Ire1 with the drug 4u8C in LPS-stimulated B cells, we saw the decrease in the short form of Xbp1 mRNA and a change in the alternative splicing pattern in the cells [48]. The total number of spliced genes shown here in Figure 2 was reduced and the ELL2 independent genes were also reduced from 3783 without drug to 2659 with 4u8C. We conclude that altering Ire1 phosphorylation with 4u8C has major repercussions for the splicing pattern of a large number of genes.

After 24 h, the activated B cell is induced to make the transcription factors Prdm1, Iref4, and ELL2, a transcription elongation factor; the cell begins a large-scale shift to produce structural changes to become an antibody-producing and -secreting factory [10,17,48]. The metabolism in antibody-secreting cells shifts to oxidative phosphorylation utilizing *NADH* and mitochondria to a larger extent than in naïve B cells [15]. By comparing ELL2 sufficient and conditionally ELL2 deficient mice for the spliced products that result 72 h after LPS stimulation, we found that ELL2 alters global splicing patterns in ASCs; see Figure 2. Having ELL2 present accounts for the splicing changes seen in ~55% of the genes in a comparison of cells plus and minus ELL2 [48] (Figure 2).

In our previous work, to elucidate its mechanism of action we added ELL2 in vitro to B cells; ELL2 was able to modulate splicing and drive first poly(A) site use in test genes, facilitated the association of polymerase associated factor and RNA processing factors with RNA polymerase II, and activated histone modifications on the *IgH* gene, especially H3K79 tri-methylation [49,50,51]. The role of ELL2 in the association of PAF and RNA processing factors to RNA polymerase was also seen in an HIV infection model [52,53]. Hence, several factors are working to change RNA processing between B cells and their progeny, the cells at 18–24 h with Ire1 phosphorylation and RIDD, and cells at 72 h after LPS stimulation that have turned on ELL2.

We had previously shown that when ELL2 was conditionally knocked out in mouse B cells those cells had distended abnormal-appearing ER, reduced amounts of the secretory-specific form of the Ig heavy chain, reduced BiP, ATF6, and cyclin B2 mRNAs, and reduced Xbp1 mRNA, both spliced and unspliced, as well as almost no transcriptionally active 40 kDa Xbp1 protein. ELL2 also has the ability to efficiently drive transcription from a canonical UPR promoter as well as the cyclin B2 promoter. We concluded that ELL2 drives secretion and at least a part of the UPR at a point after it is induced [17].

But we saw that the remaining changes in RNA processing at 72 h that did not seem to require ELL2 had already occurred early after LPS stimulation, simultaneous with RIDD activation. Thus 3783 of the 4211 changes in splicing occur early after LPS stimulation (18 h) and persist through the 72 h of stimulation, a point at which maximal Ig secretion occurs [10], as summarized in Figure 1. An additional 7361 genes are alternatively spliced in this 0–18 h interval. They do not persist until 72 h and seem to be replaced by the ELL2 dependent splicing events (Figure 2).

### 3.3. Changes in snRNA Are Correlated with the Timing of RIDD

We wondered what might be causing the changes in RNA processing we had observed at 18–24 h and focused our thinking on the ER response and the UPR in the cells. We examined the levels of the small nuclear RNAs (*snRNA*s) in B cells after LPS stimulation and saw that they fell slightly at 18–24 h post-LPS induction and fell even further at 72 h post-LPS induction, reported in [48] and summarized here in Figure 3. Next, we examined the levels of the mRNA encoding some of the transcription factors known to be important in *snRNA* synthesis. The transcription of *snRNA*s requires cooperative binding of five SnapC subunit proteins with Oct1 (and/or Oct2) to the octamer sequence for each of the *snRNA* enhancers [54,55]. A little elongation complex including Ice1 and Zc3h8 with ELL1 forms and travels with RNAPII to synthesize the *snRNA*s [56]. The Integrator (Ints) complex of at least five proteins is necessary for *snRNA* transcription termination. 

We showed that the level of mRNA for Ints 10, SnapC1 and SnapC2 are decreased after LPS stimulation [48]. This is also true for the SnapC1 (43 kDa) subunit protein. SnapC1 and 2 mRNA share the stem loop and consensus motif of other RIDD targets. When we treated B cells with 4u8C and LPS, the *snRNA*s and SnapC1 levels paradoxically increased relative to both the time zero and the 18 h plus LPS levels (see Figure 3). Thus, Ire1 phosphorylation under normal circumstances initiates events leading to a decrease of *snRNA* expression that might be expected to exert a number of the changes on mRNA splicing. We cannot ascertain if stopping the synthesis of the transcription factors for *snRNA*s is sufficient to explain their decreased levels by 72h or if there is an active process to degrade the *snRNA*s. Nuclear *snRNA*s might not be expected to be subject to RIDD in the cytoplasm like the mRNAs encoding *snRNA* transcription factor proteins. The explanation for our observations thus far remains enigmatic.

### 3.4. snRNA Levels in Cell Growth

Are changes in *snRNA* levels induced by ER stress in B cells shared with other mouse cell types and other stressors? Splenic B cells are known to be in Go and enter the S phase upon LPS stimulation within 24 h. We had previously seen that CstF64 (Cstf2), a polyadenylation factor, increases as B cells leave Go and enter the S phase just like it does in fibroblasts induced to enter the cell cycle [57]. It had previously been reported that when cells went from a state of rapid growth to stationary-phase there was a reduction in accumulation of all *snRNA*s except the 4.5S *snRNA*s. But those authors noted reductions in *snRNA*s when cells were induced to differentiate even though they were entering the S phase [58]. They offered no explanation for the decrease in *snRNA* amounts that they saw in differentiating Friend erythro-leukemia cells.

Would cells that are not professional secretory cells display RIDD-induced splicing changes? A search of the literature displays no references on RIDD and commonly used cell lines. This is an intriguing question; it should be explored to help determine if the changes we see in B cells are linked to differentiation per se, secretion, or to Ire1 activation.

## 4. Changes in *snRNA* Levels Can Alter Gene Expression

Previous studies showing that altering *snRNA*s levels could change splicing patterns include during *Drosophila* development [59], in Alzheimer’s disease where the changed levels of U1 *snRNA* lead to altered RNA processing of several mRNAs [60], and human diseases like hemato-lymphoid neoplasia, retinitis pigmentosa, and microcephalic osteodysplastic primordial dwarfism type 1 (MOPD1) associated with a loss of U4atac *snRNA* [61]. We and others have seen changes in U1A, a protein associated with U1, following B cell activation [62,63]; free U1A has been found associated with the Ig mu secretory poly(A) site where it can inhibit RNA processing [64]. U1 *snRNA* has activities outside of its role in classic intron/exon splicing; it has been shown to protect pre-mRNA from premature cleavage and polyadenylation [65,66]. Thus, changes in U1 and U1A levels could have a plethora of effects. Meanwhile, the relative enrichment of minor *snRNA*s has been hypothesized to play a crucial role in retinal neuron development [67]. Reducing the amount of the *snRNA*s following ER stress could therefore have a major effect on the splicing patterns in ASCs. The decline in *snRNA* expression and their transcription factors is visible by 18 h post-LPS and can be perturbed by an inhibitor of the RIDD activity of Ire1 [48]. The ER stress response overrides the natural tendency of the *snRNA*s to increase in response to cell growth. Some of these alterative splicing events induced early persist until the antibody-secreting stage matures, while other early splicing changes are supplanted by those introduced by ELL2.

## 5. Other Changes Associated with Stress That Could Change *snRNA*s and Splicing Patterns 

We have shown an association between falling *snRNA* levels and altered splicing but other explanations are possible for the changes in gene expression. A prime candidate for regulation is P-TEFb. CDK9 and Cyclin T1 associate to form P-TEFb, which facilitates transcription elongation and phosphorylates the Ser2 in the carboxyl-terminal domain of RNA Polymerase II to relieve pausing at viral and cellular heat shock genes [68,69]. P-TEFb can be sequestered by 7SK, LARP7 and HEXIM into an inactive state; alternatively P-TEFb can be brought into association with acetylated histones on chromatin by BRD4 where it activates transcription [70]. P-TEFb was isolated in a complex with a number of other factors, including ELL proteins in a “super elongation complex” that can drive transcription elongation and alleviate pausing [56,71]. The distribution of P-TEFb between the active and inactive complexes can be altered by a variety of cellular stresses [72,73] and perhaps by the available amount of ELL1. Since transcription elongation and RNA processing have been extensively linked [74], it is logical to predict that cellular stress and the increase in ELL2 in antibody-secreting cells might change not only the transcription profile but also the splicing patterns of a number of genes.

Another possibility for changing splicing patterns in B cell development is the change-over in expression from predominantly ELL1 and ELL3 in a B cell to that of large amounts of ELL2 in ASCs [17]. We hypothesize that this may also play a role in the splicing changes we have seen at the same time as the induction of ER stress and the UPR. ELL2 differs from the other two ELL family members [75] in the middle of the molecule where it can bind several different proteins in a yeast two-hybrid screen [76]. This could change the constellation of RNA processing factors associated with RNA polymerase II and influence splicing patterns. This has not been explored further.

The reduction in ELL1 in ASCs may have other consequences. The *snRNA* genes are thought to be transcribed by a “little elongation complex” composed of ELL1 and KIAA0947 (ICE1), NARG2 (ICE2), and ZC3H8 [56]. By having fewer ELL1 molecules available for the little elongation complex in ASCs, *snRNA* transcription could decrease and the amount of *snRNA*s would be lower by 72 h post-LPS. 

It has also been reported that p53 represses the transcription of *snRNA* genes [77] and does so by preventing the little elongation complex from forming [78]. It has been suggested that p53 dissociates ELL1 from the little elongation complex. Mdm2 is a transcription target of ELL2 [48] and it attenuates p53 function by ubiquitination [79], which might counteract the dissociation of the little elongation complex. Thus, finding *snRNA*s in such low abundance after LPS stimulation may involve the activation of several divergent pathways, some involving the ELL proteins and their pattern of expression.

## 6. Conclusions

Our findings of alternative splicing following ER induction may have major physiological and pathological implications. Thousands of genes are alternatively spliced, potentially leading to many different protein products and functional consequences. As we have described, the UPR activation is intrinsic in liver cells, pancreatic beta-cells and naïve B-cells. Beyond this, an induction of the UPR has been observed in clonal expansion of cell types such as smooth muscle cells, dysregulated calcium handling of cardiac myocytes, and protein misfolding in various brain regions. The linkage of ER stress and *snRNA* levels might be imagined to expand into disease states such as atherosclerosis, arrhythmogenesis, and neurodegenerative diseases such as Parkinson’s and Alzheimer’s disease [80,81]. 

We have discussed the importance of defining the molecular mechanisms that contribute to the complex propagation and induction of the UPR. Further investigation should be conducted into the role of RIDD and the *snRNA*s as contributors to the induction of the UPR in other cell types. Our work has implicated the UPR as a regulator of splicing working directly or indirectly through RIDD, a powerful perturbation of the status quo in cells.

## Figures and Tables

**Figure 1 ijms-19-03919-f001:**
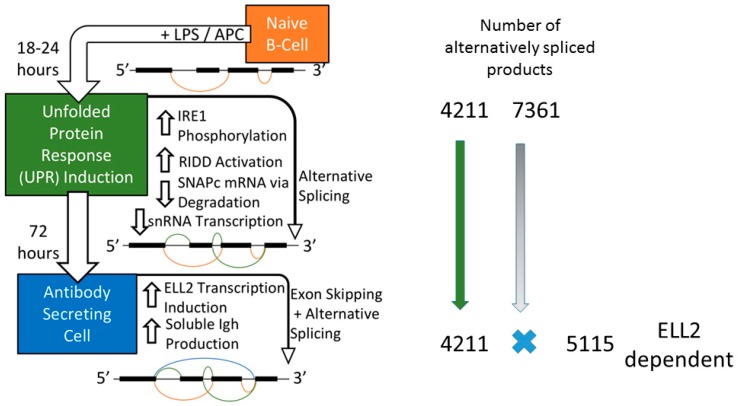
Mouse splenic B cells cultured ex vivo with lipopolysaccharide (LPS) differentiate into antibody-secreting cells (ASCs) and experience changes in splicing and gene expression. The expression of the ELL family members changes from primarily ELL1 and ELL3 in B cells to induction of large amounts of ELL2 in ASCs (not shown). The exon splicing patterns are figuratively meant to show the increasing complexity seen in cells with time. We hypothesize that the decrease of RNA for SnapC and *snRNA* is linked to the regulated Ire1-dependent mRNA decay pathway (RIDD) activity induced by endoplasmic reticulum (ER) stress. About 4211 genes spliced early (18 h) persist into the 72 h samples and are seen as ELL2 independent. Meanwhile 7361 genes alternatively spliced at 18 h are missing at 72 h and are replaced by the 5115 genes that are ELL2 dependent.

**Figure 2 ijms-19-03919-f002:**
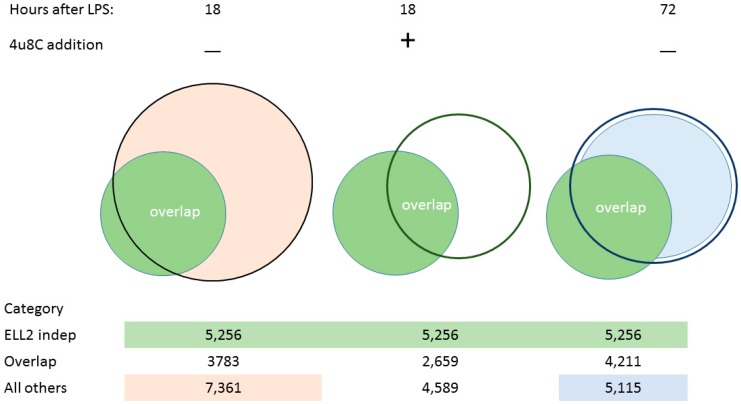
Venn diagrams of alternatively spliced genes. Left side: *ELL2* independent genes (in green and seen at 72 h) overlap with the other alternatively spliced genes at 18h in orange. Middle: Addition of 4u8C changes the splicing patterns at 18 h both reducing the number and omitting some of the ELL2 independent species. Right side: Overlap between alternatively spliced genes in the *ELL2*+/+ and the *ELL2*−/− data sets at 72 h post LPS treatment. The data associated with these diagrams are deposited in NCBI Gene Expression Omnibus under accession numbers GSE113317, GSE113475 and GSE114435.

**Figure 3 ijms-19-03919-f003:**
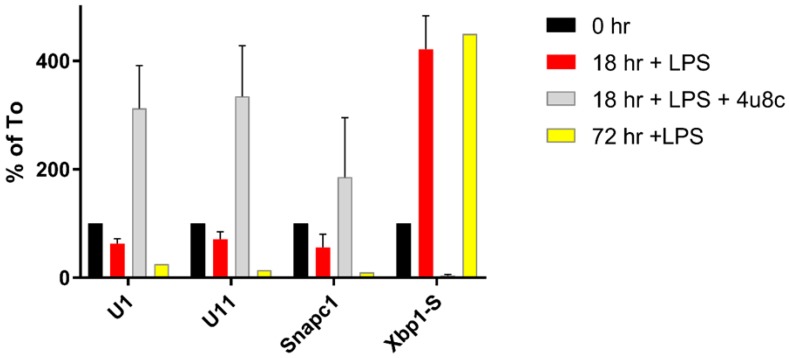
Expression of RNAs during LPS stimulation. Samples at 18 h and 72 h post-LPS induction were analyzed and compared with naïve B cells (To) set as 100%. RNA from treatment of cells with 18 h LPS and 4u8C were similarly determined relative to the To samples.

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
