# Peer review of "Linking Endoplasmic Reticular Stress and Alternative Splicing"

_ijms, 2018, doi:10.3390/ijms19123919_

Round 1

Reviewer 1 Report

This review manuscript describes the linkage between induction of alternative splicing during plasma cell differentiation and the IRE1 pathway of the mammalian endoplasmic reticulum (ER) stress response. The authors proposed that the regulated IRE1-dependent mRNA decay (RIDD) is responsible for the alternative splicing during plasma cell differentiation that they originally observed. The reviewer agrees with this novel idea, and wonders whether the RIDD directly regulates the alternative splicing or various downstream effect of the RIDD finally results in alternative splicing in plasma cells. I mean, if the RIDD directly regulates alternative splicing, alternative splicing can be observed in non-professional secretory cells such as HeLa cells in response to artificial ER stress inducers (tunicamycin or thapsigargin). If the alternative splicing does not occur in HeLa cells, the phenomenon the authors observed is specific to professional secretory cells, and it seems to be mainly regulated by differentiation, not directly by the RIDD. In that case, the title might be modified, I suppose.

<Other critiques>

(1)  Line 8: “endo-reticular stress” should be “endoplasmic reticulum stress” or “ER stress”.

(2)  Line 12: “Ire1P” should be “Ire1” because Ire1p is a yeast protein.

(3)  Line 36: “heat shock proteins” should be “ER chaperones” because all of ER chaperones (BiP, GRP94, calnexin etc.) except for HSP47 are not heat shock proteins in mammals.

(4)  Line 40: “HSP70” should be removed because BiP is not identical to HSP70 (it is a member of HSP70 family), and HSP70 is a cytosolic chaperone.

(5)  Line 41: “ER response” should be “ER stress response”.

(6)  Line 43: “heat shock proteins” should be “molecular chaperones” because some heat shock proteins are not molecular chaperones.

(7)  Lines 167-168: “inhibitor of RIDD” should be “inhibitor of Ire1”. I wonder if the authors confused RIDD with unconventional cytoplasmic splicing of XBP1 pre-mRNA. RNase activity of IRE1 regulates both of RIDD and XBP1 splicing, but they are distinct and separate processes. In mammalian cells. both are observed, while unconventional HAC1 splicing is dominant in budding yeast and only RIDD is observed in fission yeast.

(8)  Line 275: The reviewer wonders if increased degradation of snRNAs is irresponsible for decreased expression of snRNAs and alternative splicing during plasma cell differentiation. Is it possible?

(9)  Line 353: Reference 7 is not appropriate because it is published in a method journal. The authors should cite appropriate references, that is, original papers reporting XBP1 splicing (Calfon et al., Nature 2002 and Yoshida et al., Cell 2001).

Author Response

Reviewer #1 makes an excellent point regarding whether non-professional secreting cells would display RIDD induced splicing changes.  We added this idea to our discussion in red text after describing the snRNA decay.  Thank you for helping us improve the document.

1.      We changed endo-reticular stress to endoplasmic reticulum stress in line 8 and ER stress elsewhere.

2.      Changed Ire1 P to phosphorylated Ire1

3.      Heat shock proteins changed to ER chaperones as requested

4.      Removed HSP70 line 40

5.      Line 41 changed to ER stress response

6.      Line 43 changed to molecular chaperones

7.      Changed lines 167-8 to inhibitor of Ire1.  Thank you for the insight that HAC1 splicing is dominant in budding yeast while only RIDD is observed in fission yeast.  We added a short paragraph in red text and several references about this.

8.      Line 275 “The reviewer wonders if increased degradation of snRNAs is irresponsible (sic) for decreased expression of snRNAs and alternative splicing during plasma cell differentiation. Is it possible?”  If the question is, “is it responsible?” then we have discussed this in the paragraphs on how decreased snRNAs can klead to altered splicing and if irresponsible we mention other possibilities like the decrease in ELL1 as contributing to decreased snRNA levels.  I hope we have covered both possibilities.

9.      Line 353 / references we added Calfon et and Yoshida et al references to the text and reference list. Thank you for pointing out that we should have cited original not secondary sources.

Reviewer 2 Report

 The review entitled "Linking endoplasmic reticular stress and alternative splicing" by Nolan T. Carew et al have investigated ER stress, specifically the impact of RIDD on the pattern of mRNA expression, particularly on B cells. This is a thorough and well-written study based on findings which are published in high impact factor journals and authors’ own findings. 

Minor Comments:

1.     The typo mistake at the page of 2 line 71.

2.     The typo mistake at page 1, line 8-9: Endoplasmic reticular stress instead of endo-reticular stress.

3.     Page 4 section 2.4. The reviewer is curious if there is any relevant study with 4PBA on MIN6 (pancreatic beta cell line) cells to establish the opposite action of thapsigargin (TG).  

Author Response

Reviewer # 2:

1.      Typo p 2 line 71 not sure what was meant.

2.      Page 1 line8-9 typo changed to endoplasmic reticular stress as also suggested by reviewer #1

3.      We have preliminary data on the MIN6 line and TG showing decreased snRNAs with stress. A previous editor told us not to mix data and a review so we will publish that data elsewhere.

Round 2

Reviewer 1 Report

Since the authors completely improved the manuscript, the reviewer thinks that the manuscript is now ready for publication in the journal of IJMS.

Author Response

thank you. I added one paper I found in my search prompted by reviewer #1s comments. highlighted red in text

Reviewer 2 Report

 The reviewer thinks a period will be applied after secretion on page 2, line 71.

Author Response

added period page 2 line 71.